# Characterization of the Cardiac Structure and Function of Conscious D2.B10-*Dmd^mdx^*/J (D2-*mdx*) mice from 16–17 to 24–25 Weeks of Age

**DOI:** 10.3390/ijms241411805

**Published:** 2023-07-22

**Authors:** Daria De Giorgio, Deborah Novelli, Francesca Motta, Marianna Cerrato, Davide Olivari, Annasimon Salama, Francesca Fumagalli, Roberto Latini, Lidia Staszewsky, Luca Crippa, Christian Steinkühler, Simonetta Andrea Licandro

**Affiliations:** 1Department of Cardiovascular Medicine, Istituto di Ricerche Farmacologiche Mario Negri IRCCS, 20156 Milan, Italy; daria.degiorgio@marionegri.it (D.D.G.); deborah.novelli@marionegri.it (D.N.); francesca.motta@marionegri.it (F.M.); marianna.cerrato@marionegri.it (M.C.); olivari.davide@gmail.com (D.O.); annasimon.salama226@gmail.com (A.S.); francesca.fumagalli@marionegri.it (F.F.); roberto.latini@marionegri.it (R.L.); lidia.staszewsky@marionegri.it (L.S.); 2School of Medicine and Surgery, University of Milano-Bicocca, 20126 Milan, Italy; luca.crippa@unimib.it; 3New Drug Incubator Division, Italfarmaco S.p.A., 20126 Milan, Italy; c.steinkuhler@italfarmacogroup.com

**Keywords:** Duchenne muscular dystrophy, cardiomyopathy, D2-*mdx* mice, echocardiography, cardiac circulating biomarkers, fibrosis

## Abstract

Duchenne muscular dystrophy (DMD) is the most common form of muscle degenerative hereditary disease. Muscular replacement by fibrosis and calcification are the principal causes of progressive and severe musculoskeletal, respiratory, and cardiac dysfunction. To date, the D2.B10-*Dmd^mdx^*/J (D2-*mdx*) model is proposed as the closest to DMD, but the results are controversial. In this study, the cardiac structure and function was characterized in D2-*mdx* mice from 16–17 up to 24–25 weeks of age. Echocardiographic assessment in conscious mice, gross pathology, and histological and cardiac biomarker analyses were performed. At 16–17 weeks of age, D2-*mdx* mice presented mild left ventricular function impairment and increased pulmonary vascular resistance. Cardiac fibrosis was more extended in the right ventricle, principally on the epicardium. In 24–25-week-old D2-*mdx* mice, functional and structural alterations increased but with large individual variation. High-sensitivity cardiac Troponin T, but not N-terminal pro-atrial natriuretic peptide, plasma levels were increased. In conclusion, left ventricle remodeling was mild to moderate in both young and adult mice. We confirmed that right ventricle epicardial fibrosis is the most outstanding finding in D2-*mdx* mice. Further long-term studies are needed to evaluate whether this mouse model can also be considered a model of DMD cardiomyopathy.

## 1. Introduction

Duchenne muscular dystrophy (DMD) is the most common form of muscle degenerative disease. It is an X-linked disorder caused by mutations in the dystrophin-encoding gene, which leads to the loss of a functional dystrophin protein [1]. The estimated incidence of DMD is 1 in 5000 live male births [2]. This disease is characterized by a progressive loss of muscle mass due to degeneration and necrosis, with a subsequent muscle replacement by fibrotic and adipose tissue and calcifications [3]. These muscular changes are the reasons for the progressive weakness and wasting of both skeletal muscles and myocardium.

D2.B10-*Dmd^mdx^*/J mice (D2-*mdx*) represents a DMD murine model generated by backcrossing *mdx* mice onto the D2.B10/J parent strain, transferring the spontaneous point mutation in exon 23 of the dystrophin gene from *mdx* mice to the D2.B10/J background [4]. This recently developed murine model shows higher fibrotic tissue infiltration in both cardiac and skeletal muscles due to an excessive release of the transforming growth factor-β (TGF-β) by the latent TGF-β binding protein (Ltbp) 4 during the persistent inflammatory process that occurs in dystrophic muscles after a chronic injury [5]. The *LTBP4* gene is known to influence the loss of ambulation in DMD patients, and the most severe form of the disease occurs in patients homozygous for the VTTT haplotype of Ltbp4 [6]. D2-*mdx* mice functionally mimic this human haplotype [7,8]. Van Putten and colleagues and, more recently, Hayes and colleagues already studied fibrosis in D2-*mdx* mice at different ages (10–34 weeks and 10–18 weeks, respectively), pointing out an increase in interstitial fibrosis with age. Although mean levels increased over time, these differences did not reach statistical significance due to a large individual variation [4,9]. Hayes and colleagues found that fibrosis was mainly distributed on the epicardium of the right ventricle (RV) in the absence of structural and functional left ventricle (LV) impairment [9]. Cooley and colleagues found diaphragmatic and cardiac fibrosis and calcification in 7-, 28-, and 52-week-old D2-*mdx* mice, although the infiltrate was not measured. In this last study, LV systolic function was normal at seven weeks of age and significantly decreased starting from 28 up to 52 weeks of age [10].

To date, there is no curative therapy for DMD patients; however, glucocorticoid (GC) steroid treatment, corrective orthopedic surgery, and assisted ventilation can contribute to improving the quality of life of patients and to delaying disease progression. Recent developments have seen the approval of exon-skipping oligonucleotides (Eteplirsen, Golodirsen, and Vitolarsen) by the FDA [11,12,13] and Ataluren for patients with nonsense mutations by EMA [14,15]. Only a minority of patients can benefit from these treatments and the mainstay of disease management is GC steroids. All of these treatments target early-stage patients with the aim of prolonging ambulation. In their later life, DMD patients experience respiratory and cardiac sequels. The progression of cardiac manifestations such as dilated cardiomyopathy (DCM) has become one of the primary causes of death of DMD patients [3,16].

Preclinical signs of cardiac manifestations in DMD patients can be detected starting from six years of age up to 10 years and DCM incidence increases with age. By the age of 18 years, 70% of DMD patients show overt DCM that progresses to heart failure and subsequent death of the patient in the terminal stage of the disease [17]. Dysfunction or lack of dystrophin leads to instability of the membranes of the cardiac muscle fibers, leading to cardiomyocyte death. Fibroblasts invade the damaged area and form scar tissue (fibrosis) in the heart and the increase in LV interstitial fibrosis reduces the efficiency of myocardial function [18].

In order to identify the appropriate age for starting a new preventive or curative antifibrotic treatment, overcoming the controversies reported in the literature, we attempted to comprehensively characterize the cardiac structure and function in conscious D2-*mdx* mice by assessing left ventricular function and cardiac fibrosis. The evolution of cardiac manifestations in D2-*mdx* mice was followed by echocardiography for eight weeks, starting from the age of 16–17 weeks and comparing the data to age-matched healthy mice. In order to provide a more accurate description of cardiac anatomical damage and functional impairment, histological analyses and biomarker assays were performed in a subgroup of 16–17-week-old mice and at the end of the experiment, when the mice were 24–25 weeks old. Additionally, fibrosis infiltrate in the diaphragm muscle was also evaluated in 24–25-week-old mice to assess DMD progression in skeletal muscle.

## 2. Results

### 2.1. Survival and Body Weight

At baseline, the mice were 16–17 weeks old. All animals survived for eight weeks until the end of the experiment. Body weight (BW) in the DBA/2J wild-type (WT) mice was significantly higher compared to that in the D2-*mdx* mice (*p* < 0.0001 at all time points) (Figure 1). When mice were 16–17 weeks old, BW was 19.8% lower in the D2-*mdx* mice compared to the DBA/2J WT mice; this difference slightly increased at the end of the follow-up (−22.8%).

### 2.2. Left Ventricular Structure and Function Evaluated by Echocardiography in 16–17-Week-Old Mice

During the echocardiographic examination, hyper-echogenic areas were observed in the D2-*mdx* mice and, in minor extension, in some of the DBA/2J WT mice. These images were predominantly identified in the surrounding epicardium of both the RV and LV, in the LV wall (Figure 2 and Appendix A), papillary muscles, mitral valvular, and sub-valvular apparatus and in pulmonary and aortic leaflets. The mean values of the echocardiographic variables of the 16–17-week-old DBA/2J WT and D2-*mdx* mice are described in Appendix A. The heart rate (HR) and left ventricular ejection fraction (LVEF) were 11.1% and 8.8% lower in the D2-*mdx* mice compared to the DBA/2J WT mice, respectively (Figure 3A,B). The same trend was observed for cardiac output (CO), stroke volume (SV), and mitral E inflow velocity (E vel) (−32.6%, −21.5%, and −31.5% respectively (Figure 3C–E). The pulmonary artery acceleration time (PAAT) resulted significantly lower in the D2-*mdx* mice compared to the DBA/2J WT mice (−25.6% and −33%, respectively) (Figure 3F,G).

### 2.3. Left Ventricular Structure and Function Evaluated by Echocardiography in 24–25-Week-Old Mice

HR was significantly higher in the DBA2/J WT mice compared to the D2-*mdx* mice at all time points considered for echocardiographic examination (+15% at 20–21 weeks of age; +11% at 24–25 weeks of age) (Figure 4A).

LVEF was lower in the D2-*mdx* mice compared to the DBA2/J WT mice at each time point, but the difference was statistically significant only when the mice reached 24–25 weeks of age (∆ = 17%) (Figure 4B). Interestingly, 5/10 (50%) of the D2-*mdx* mice showed a ten percent point decrease in LVEF, considering changes between 16–17 weeks and 24–25 weeks of age. Left ventricle end-diastolic volume indexed by body weight (LVEDV/BW) was higher in the DBA/2J WT mice than in the D2-*mdx* mice (*p* > 0.05) and increased over time in both groups (Appendix A). From 16–17 to 20–21 weeks of age, left ventricle end-systolic volume indexed by body weight (LVESV/BW) values were the same for both DBA/2J WT and D2-*mdx* mice. At 24–25 weeks of age, LVESV/BW increased in the D2-*mdx* mice compared to the DBA2/J WT mice, but the difference was not statistically significant (*p* = 0.07) (Appendix A). No significant differences in anterior or posterior diastolic wall thickness (AWTh_d and PWTh_d, respectively) were found between the D2-*mdx* and DBA/2J WT mice at any time point (Appendix A).

E vel and A vel were significantly lower in the D2-*mdx* mice compared to the DBA/2J WT mice at all time points. A 32% lower E vel was observed in the D2-*mdx* mice compared to the DBA/2J WT mice at 16–17 weeks of age and this difference was maintained until the end of the study (−31% and −29% at 20–21 and 24–25 weeks of age, respectively) (Figure 4C). Similar results were observed for A vel (−31%, −27%, and −26% at 16–17, 20–21, and 24–25 weeks of age, respectively; Figure 4D). CO and SV were significantly lower in the D2-*mdx* mice compared to the DBA/2J WT mice (CO: −33% and −41% at 16–17 and 24–25 weeks of age, respectively; SV: −22% and −30% at 16–17 and 24–25 weeks of age, respectively; Figure 4E,F). PAAT and ET were also significantly lower in the 24–25-week-old D2-*mdx* mice compared to the DBA/2J WT mice (−20% and −7%, respectively) (Figure 4G,H). RV CO and SV were significantly lower in the D2-*mdx* mice compared to the DBA/2J WT mice (CO: −25% and −49% at 20–21 and 24–25 weeks of age, respectively; SV: −26% and −45% at 20–21 and 24–25 weeks of age, respectively) (Appendix A).

### 2.4. Left and Right Ventricular Epicardial and Interstitial Fibrocalcinosis in 16–17- and 24–25-Week-Old Mice

#### 2.4.1. Gross Pathology

RV + LV weight on BW was higher in the D2-*mdx* group when compared to the DBA/2J WT mice starting from 16–17 up to 24–25 weeks of age (+10.7% and +7.5%, respectively) (Figure 5). The pericardium of both ventricles from the D2-*mdx* mice showed white thickened and irregular areas mainly distributed on the RV. The DBA/2J WT mice also presented small white areas on the RV epicardium (Figure 2). The extension of RV crusts in the D2-*mdx* mice was 8.2-fold larger compared to in the DBA/2J WT mice (Figure 6A). Crusts on the LV covered a significant lower area (29 ± 3.9%) compared to that on the RV (73.4 ± 4%) (Figure 6B). Crust thickness was similar in both ventricles of the D2-*mdx* mice (LV: 0.2 ± 0.02 mm; RV: 0.2 ± 0.01 mm) but significantly higher compared to the DBA/2J WT (Figure 6C,D).

#### 2.4.2. Histological Analysis

Histological evaluation in the 16–17-week-old DBA/2J WT mice showed the absence of LV fibrosis (Figure 7A). In the D2-*mdx* mice, LV epicardial fibrosis was significantly higher compared to interstitial fibrosis (5.6 ± 1.0% and 0.8 ± 0.2%, respectively). The 24–25-week-old mice showed +55.6% LV interstitial myocardial fibrosis and −69.7% epicardial fibrosis compared to the younger mice; consequently, total LV fibrosis was −25.5%, and all of these differences were statistically significant (Figure 7B–D). The intraventricular septum (IVS) fibrosis percentage in the 16–17-week-old D2-*mdx* mice was comparable to that in the 24–25-week-old D2-*mdx* mice (1.4 ± 0.7% and 2.1 ± 0.8%, respectively), and higher when compared to the DBA/2J WT mice (*p* < 0.05 only for the 16–17-week-old mice) (Figure 7E).

At 16–17 weeks of age, a higher percentage of epicardial fibrosis was found in the RV (31.7 ± 4.6%) compared to interstitial fibrosis (1.4 ± 0.4%). In the 24–25-week-old mice, the percentage of RV fibrosis was higher when compared to that of younger mice, but the differences were not statistically significant due to a high variability or to the absence of fibrosis in some mice (Figure 7F–H).

### 2.5. Plasma Concentration of Cardiac Biomarkers in the 16–17- and 24–25-Week-Old Mice

The high-sensitivity cardiac troponin T (Hs-cTnT) median plasma concentration in the 16–17-week-old D2-*mdx* mice was two-fold higher compared to that in the DBA/2J WT mice (median (range): 103 (68.5–9399) ng/L and 45.3 (12.2–434.1) ng/L, respectively) (Figure 8A). While the young D2-*mdx* mice did not present a higher plasma concentration of Hs-cTnT except for two mice who may have already developed the pathology, the D2-*mdx* adult mice showed a robust three-fold increase in Hs-cTnT plasma concentration compared to the DBA/2J WT mice (median (range): 389 (82–660) ng/L and 102 (26–379) ng/L, respectively) (Figure 8A). N-terminal pro-atrial natriuretic peptide (NT-proANP) levels were not statistically significant different between the D2-*mdx* and WT mice (median (range): 6.9 (2.8–13.5) ng/mL and 6.6 (2.5–10.1) ng/mL, respectively) (Figure 8B).

### 2.6. Left and Right Ventricles and Intraventricular Septum Interstitial CD45+ Cell Analysis in 24–25-Week-Old Mice

The CD45+ cell density in both the D2-*mdx* and DBA/2J WT mice was close to the upper limit of normality (arbitrary level of normality, 14 cells/mm^2^). Cardiac leukocyte infiltration was higher in the RV, even though with a higher variability, LV, and IVS of the D2-*mdx* mice compared to the DBA2/J WT mice, and in all cases, the results were not statistically significant (Appendix A).

### 2.7. Fibrosis Quantification in the Diaphragm of 24–25-Week-Old Mice

The diaphragm of the DBA2/J WT mice showed a low percentage of fibrotic areas when compared to D2-*mdx* muscles and the difference was statistically significant (1.6 ± 0.3% and 21.7 ± 3.4%, respectively) (Figure 9).

## 3. Discussion

The results from this study highlight that the D2-*mdx* mouse model presents some characteristics of the DMD disease in humans, specifically on the heart on both functional and histopathological levels, in young and adult mice. A decrease in LV contractile function together with an increase in myocardial interstitial fibrosis was observed in both ventricles. To add significance to these results, increased plasma levels of hs-cTnT in the D2-*mdx* mice were observed.

DMD murine models, however, also show different limitations in mimicking human cardiac physiopathological damage and its progression. The D2-*mdx* strain was proposed as an alternative to the classical *mdx* murine model on C57BL/10ScSn background, in which myocardial impairment is much less pronounced compared to human disease and develops only in adults. Coley and colleagues pointed out that an increase in myocardial interstitial fibrosis in D2-*mdx* mice leads to a significant impairment of LV function [10], while, more recently, Hayes and colleagues claimed the opposite [9].

In view of these contradictions, the aim of this study was to characterize the overall cardiac function and structure by both in vivo and ex vivo analyses to determine possible heart damage and its evolution to cardiomyopathy over time in D2-*mdx* mice from 16–17 to 24–25 weeks of age. These evaluations were performed by different methods, some of them never used before. For the first time, the evaluation of LV structure and function was performed by echocardiography in conscious mice; these data were supported by the plasma levels of circulating cardiac biomarkers. These analyses were accompanied by ex vivo morphometric analyses of both the heart and diaphragm muscles and immunohistochemical analyses of the heart to evaluate inflammatory infiltrate.

Transthoracic echocardiography is an inexpensive and non-invasive imaging technique frequently used in many experimental settings for studying DMD mouse models [9] and may be carried out in conscious mice, avoiding the echocardiographic alterations induced by anesthesia in heart rate, chamber dimension, and systolic function [19]. However, echocardiography in conscious mice needs to be followed by two operators, since one handles the animal with one hand and the probe with the other hand, while the second operator works with the echo machine.

In this study, we observed that at 16–17 weeks of age, LV systolic function parameters such as LVEF, CO, and SV were mildly but significantly compromised in the D2-*mdx* mice compared to the DBA/2J WT mice. Moreover, a decrease in the E vel and A vel parameters was observed. During the eight weeks of follow-up (until 24–25 weeks of age), a significant worsening of LV systolic function was observed. A negative correlation between LVEF and LV interstitial fibrosis in the D2-*mdx* mice at 24–25 weeks of age was found, meaning that those mice with lower LVEF had a greater extension of LV fibrosis (Appendix A). Our data showed no differences in LV wall thickness, as also described by Hayes and colleagues in mice [9] and by Puchalski and colleagues in patients [20].

Variables such as PAAT and PAAT/ET were significantly impaired in both the young and adult D2-*mdx* mice. PAAT and ET are considered useful variables for the indirect evaluation of pulmonary vascular resistance; they are directly associated with RV function and indirectly correlated to pulmonary systolic pressure. An increase in RV fibrosis leads to RV remodeling and subsequently RV dysfunction with a low CO and SV. It could be assumed that these changes, in addition to the high fibrotic infiltration in the diaphragm muscle (responsible for respiratory dysfunction and consequent hypoxia), cause increased pulmonary vascular resistance. In fact, we found a negative correlation between PAAT and RV total fibrosis in the D2-*mdx* mice at 16–17 weeks of age (Appendix A). This may suggest that those mice with low a PAAT had a greater extension of total fibrotic percentage in the RV. A similar phenomenon was also observed in the D2-*mdx* mice at the end of the follow-up, even though no significant correlation was found due to the high inter-individual variability.

Ventricular weight normalized for body weight was significantly increased in the D2-*mdx* mice of 16–17 weeks of age (+8%) and the D2-*mdx* mice of 24–25 weeks of age (12%) compared to the respective DBA/2J WT mice, probably due to the neoformations of fibrocalcinosis found on the pericardium of both ventricles.

At autopsy, the pericardium of the D2-*mdx* mice appeared with white, irregular, and thickened areas, mainly located on the RV; this phenomenon was observed in both the young and adult D2-*mdx* mice. This outer layer is characterized by minimal foci of fibrocalcinosis, defined “crusts” by Eaton and colleagues [21].

Fibrosis is the extent of collagen measured by Sirius Red staining. Histopathological analyses in the heart allowed to differentiate the percentage of fibrotic areas according to the location (epicardial or interstitial) in both the RV and LV. We observed that in both the young and adult D2-*mdx* mice, fibrosis was 5.5 times more extended compared to in the DBA/2J WT mice and the RV epicardial fibrosis was 30–35% larger compared to interstitial fibrosis. The mean values of RV interstitial fibrosis increased in the adult D2-*mdx* mice compared to the younger mice; this difference was not significant due to the high interindividual variability. LV and IVS interstitial fibrosis remained stable. Furthermore, as already described by Van Putten and colleagues, a marked individual variation in the percentage of both epicardial and interstitial fibrosis in the D2-*mdx* mice was observed [4]. Differently from DMD patients, our results highlighted a greater extension of fibrotic areas in the RV epicardium, as previously described by Hayes and colleagues [9], as well as by other authors [4,10].

The myocardial tissue changes underlying cardiac dysfunction in DMD disease are well described and refer to a progressive replacement of damaged cardiomyocytes with adipose and fibrotic tissue until cardiomyopathy develops [18]. In DMD patients, cardiac magnetic resonance imaging (cMRI) revealed multifocal subepicardial fibrosis, mainly localized in the LV free wall [22]. A postmortem study of three DMD young male patients revealed patches of interstitial fibrosis also in the papillary muscles [23]. In contrast, the mitral valve, its leaflets, and chordae tendineae were completely normal [23], a finding that differs from what we observed in the D2-*mdx* mice. Indeed, in our study, hyper-echogenic areas were observed in both the young and adult D2-*mdx* mice, mainly at the level of the valves, the interventricular septum, the sub-valvular mitral apparatus, and surrounding the RV pericardium. Hyper-echogenic areas were also observed in the LV wall, papillary muscles, mitral annulus, and tendinous chordae, in the mitral, pulmonary, and aortic leaflets, and also in the myocardium, without observing or finding abnormalities in the LV wall thickness. Puchalski and colleagues also observed significant regional abnormalities of the wall motion on gadolinium-enhanced cMRI and evidenced a reduction in wall thickening at fibrotic sites [20].

To complement the in vivo assessment of myocardial damage and increased stress walls, cardiac plasma biomarkers such as hs-cTnT and NT-proANP were considered to provide translational value to the study. Indeed, biomarker studies in the blood of DMD patients are rapidly expanding [24,25,26]. In our study, we observed hs-cTnT to be significantly higher in the D2-*mdx* mice than in the DBA/2J WT mice at 24–25 weeks of age, and in some mice already at 16–17 weeks of age, indicating cardiomyocyte damage even at this relatively early stage. It is important to point out that increased hs-cTnT levels were measured at 16–17 weeks of age in only two D2-*mdx* mice, while this occurred in 70% of mice at 24–25 weeks of age. The NT-proANP plasma concentration at 24–25 weeks of age was not significantly different between the D2-*mdx* and DBA/2J WT mice; this finding is explained by the mild-to-moderate impairment of LV function observed in the D2-*mdx* mice.

In DMD patients, cardiac troponin elevations are poorly understood and have no clear significance; indeed, it is still debated whether troponin is only a marker of heart damage or also of skeletal muscle and kidney damage, fractures, etc. [27].

In the same mice, we performed immunohistochemical analyses, investigating the activation of inflammation through the identification of CD45+ cells. The results of the CD45+ cell density are consistent with those related to a low level of interstitial fibrosis, suggesting a low inflammatory activation in the D2-*mdx* mice at 24–25 weeks of age.

In conclusion, anatomical and functional pathological manifestations in both the LV and RV varied from mild to moderate in the D2-*mdx* mice compared to the DBA2/J WT mice at all time points analyzed in this study. The following limitations should be considered. Since the objective of our work was to evaluate the function and structure of the LV, evaluation of the respiratory function was not considered information that could be useful for the interpretation of echocardiographic results related to the pulmonary resistance parameters.

## 4. Materials and Methods

### 4.1. Study Approval

Procedures involving animals and their care were carried out in conformity with institutional guidelines in compliance with national and international laws and policies (Italian Governing Law: D.lgs 26/2014 “Attuazione della direttiva 2010/63/UE sulla protezione degli animali utilizzati a fini scientifici”). The research project was authorized by the Italian Ministry of Health (n° 187/2018-PR).

### 4.2. Animals

The mice were kept under pathogen-free conditions with a 12 h light/12 h dark cycle at a temperature of 22° ± 2° and 55% ± 10% humidity. Each cage was enriched with a mouse house and nesting material. The mice were regularly checked by a certified veterinarian who was responsible for animal welfare supervision, health monitoring, experimental protocols, and procedure revision. During the follow-up study, the mice received food (VRF1 diet, Charles River) and water ad libitum. For the study, DBA/2J WT healthy (stock No: 625) and D2.B10-*Dmd^mdx^*/J (D2-*mdx*) (stock No: 013141) 6–7-week-old male mice were purchased from Charles River (France) and The Jackson Laboratories (Bar Harbor, ME, USA), respectively. All mice were grown up to 16–17 weeks of age and then their body weight and echocardiographic parameters were recorded.

### 4.3. Study Design

After the first echocardiographic exam in 40 mice (20 DBA/2J WT and 20 D2-*mdx* mice), a sub-group of 20 animals (10 DBA/2J WT and 10 D2-*mdx* mice) were immediately euthanized for histological and cardiac biomarker analyses. The remaining 20 mice were followed for eight weeks. BW was evaluated weekly, while echocardiographic examinations were performed every four weeks until the animals were 24–25 weeks old. Afterward, the mice were immediately euthanized for histological and cardiac biomarker analyses (Figure 10).

### 4.4. Echocardiography

Transthoracic echocardiography (ARIETTA V70—FUJIFILM Healthcare, Twinsburg, OH, USA) was performed in conscious mice at Week 0 (16–17-week-old mice), Week 4 (20–21-week-old mice), and Week 8 (24–25-week-old mice) from the start of the experiment. Each mouse underwent echocardiographic examination after five days of training (5 min/mouse, once a day) mimicking echocardiographic assessment. During the training, the sonographer handled the mice in supine position and, after the application of pre-warmed echo transmission gel, the echocardiographic probe was passed over the thoracic region; mouse body temperature was maintained using an infrared lamp that provided supplemental heat. A 18 MHz linear transducer at high frame rate imaging (110 Hz) and a 13 MHz phased array probe for Doppler (pulsed-wave, continuous, color) and tissue Doppler analysis were used. Pulmonary vascular resistance was assessed by measuring the pulmonary artery acceleration time (PAAT) and the ejection time (ET) from the parasternal short-axis view at the level of the aortic valve. Pulsed-wave Doppler recording of the pulmonary blood flow was obtained by placing the sample volume at the tip of the pulmonary valve leaflets. A visual assessment of the wave shape and the measure of the PAAT was performed. To account for heart rate variability, the PAAT was adjusted for ET (PAAT/ET). ET was measured as the time from the closure and the opening of the pulmonary valve; in other words, it is the time interval between the onset of flow and peak transpulmonary valvular flow; an increase in this interval reflects the difficulties of RV emptying (for example, secondary to pulmonary arterial hypertension or pulmonary valvular stenosis). The LV volumes (end diastolic volume, EDV, end systolic volume, ESV) and left ventricular ejection fraction (LVEF) were calculated by the modified simple plane Simpson’s rule from the parasternal long-axis view as previously reported. Parasternal long-axis and four and five apical chamber views were used for 2D and color flow imaging and spectral Doppler study of the mitral valve and/or aortic outflow tract. The LV stroke volume (SV), cardiac output (CO), mitral inflow velocity (E vel) and diastolic function parameters were obtained and calculated. All recordings and measurements were followed according to the recommendations of the American and European Societies of Echocardiography Guidelines [28,29] as reported in our previous works [30]. Echocardiographic recordings were saved on a USB external hard disk for offline analysis by a cardiologist blinded to the study groups.

### 4.5. Blood Sampling and Troponin and N-Terminal Pro-Atrial Natriuretic Peptide Assay

After each echocardiographic evaluation, the mice were kept under deep anesthesia with isoflurane and blood samples were drawn from the inferior cava vein. The blood was kept in EDTA K2 tubes and immediately centrifuged, and the plasma was then aliquoted (200 µL) and stored at −80 °C for subsequent biomarker assays. To assess the myocardial injury, high-sensitivity cardiac troponin T (hs-cTnT) was measured at 16–17 and 24–25 weeks with an electrochemiluminescence assay (ECLIA, Roche Diagnostics, Mannheim, Germany), while N-terminal pro-atrial natriuretic peptide (NT-proANP), a marker of myocardial wall stress [31], was assayed with an ELISA kit (Biomedica, Wien, Austria, BI-20892) following the manufacturer’s recommendations [32].

### 4.6. Heart Fixation

After blood sampling, each mouse was immediately euthanized by a 2.5 M KCl intravenous injection. The heart was excised with careful dissection from the surrounding tissues, and the atria were separated from the ventricles and the ventricles weighed. In the 16–17-week-old mice, the heart was preserved in 10% buffered formalin for 24 h and then scanned for macro-histological analyses. In the 24–25-week-old mice, the ventricles were divided into two cross-sections: the basal fixed in Optimal Cutting Temperature (OCT) compound and the apical fixed in 10% buffered formalin by immersion. The formalin-fixed samples were then embedded in paraffin for subsequent histological analyses.

### 4.7. Macro-Histological Analyses

Digitized images of the anterior and posterior sides of the formalin-fixed hearts of 16–17-week-old mice were taken using a high-resolution scanner (Epson perfection 3200 Photo, Los Alamitos, CA, US). Afterward, the ventricles were divided into three coronal slices (basal, median, and apical) of 2–3 mm by a heart matrix. Digitized images of the apical and basal sides of previously described sections were taken. Thus, it was possible to estimate the extension of epicardial fibrosis/calcification by evaluating the angular extension and thickness in both RV and LV by Image J (1.47v, WayneRasband, National Institutes of Health, Bethesda, MD, USA). Extension of epicardial fibrosis/calcification was calculated on basal, median, and apical coronal slices expressed as the percentage of calcified region on the extension of the RV or LV. The calcification thickness was measured at least in three points and expressed in millimeters as the extent of both RV and LV epicardial calcification (Figure 6A2,3). All measurements were carried out by two investigators blinded to experimental groups and mean values were considered.

### 4.8. Histological Analyses

A first assessment of the myocardial morphology on 5 μm sections was evaluated by Hematoxylin–Eosin staining. The collagen content was measured in 10 μm of 0.1% Sirius Red-stained sections. Epicardial fibrosis is defined as the external layer of collagen surrounding both ventricles, while interstitial fibrosis as the percentage of collagen located inside the myocardium. Both epicardial and interstitial collagen are expressed as the percentage of collagen area in the RV, IVS, and LV (i.e., LV collagen area/total LV area). The nature of the Sirius Red-stained collagen deposit was confirmed by examining the sections under a microscope fitted with a linear polarizing filter that renders collagen fibers birefringent. The measurements were carried out by two investigators blinded to experimental groups and mean values were considered.

### 4.9. CD45+ Cell Immunostaining

The interstitial inflammatory response was assessed at T8, and infiltrating leukocytes were detected by anti-CD45 immunohistochemical staining. Basal heart cryosections (10 μm) were incubated with a monoclonal rat anti-mouse CD45 antibody (Biotin Rat Anti-mouse CD45 (30-f11) IgG, 553078, BD Pharmingen, San Jose, CA, USA). Peroxidase activity was visualized by diaminobenzidine (DAB Substrate Kit, Peroxidase (HRP), with Nickel, (3,3-diaminobenzidine), room temperature). CD45 cells were identified and quantified in the myocardium of the LV, RV, and IVS.

### 4.10. Skeletal Muscle Tissue Preparation, Sectioning, and Staining

Diaphragm muscle was collected from sacrificed DBA/2J WT and D2-*mdx* mice at the end of the follow-up to be examined by a pathologist. The diaphragm was fixed in 10% buffered formalin solution (Bio-optica, Milan, Italy) at +4 °C for at least 48 h. After fixation, the muscular samples were transversely trimmed, caged, and paraffin-embedded overnight with an Automated Vacuum Tissue Processor Floor (ETP, Histo-Line Laboratories, Pantigliate, Italy) and included in paraffin blocks. Serial transverse cross-sections (4 µm thick) were cut with a microtome (Leica RM 2255, Leica, Wetzlar, Germany), collected onto uncoated glass slides, and stained using the standard protocol for Sirius Red staining (Direct Red 80, Sigma-Aldrich, St. Louis, MO, USA). The fibrotic area, corresponding to the area stained in red, was compared to the total area of the tissue section, and the results are expressed as the percentage of fibrosis. The percentage of fibrosis of muscles in the different experimental groups was expressed by averaging the three values obtained from each muscle.

### 4.11. Image Acquisition and Fibrosis Quantification in Both the Heart and Skeletal Muscle

Entire heart sections were acquired at 20× with a pixel size of 0.346 μm by an Olympus BX-61 Virtual Stage microscope equipped with a motorized platform (Olympus, Hamburg, Germany) and digitized. Acquisition was done over 5 μm thick stacks, with a step size of 1 μm. For skeletal muscle fibrosis quantification, Sirius Red-stained slides were examined by an Olympus BX-51 light microscope. From each sample, one to three random microphotographs at a magnification of 20× were collected using the Image-Pro Plus system. The digital images were processed by a pathologist using ImageJ software 2.9.0 (U.S. National Institutes of Health, Bethesda, MD, USA).

### 4.12. Statistical Analysis

All values are expressed as the mean ± standard error (SEM), except for hsTnT and NT-proANP, which are expressed as median and Q1-Q3. The Shapiro Wilk test for normality was executed for all variables. The number of mice per group might change according to the measurements performed. Two-way ANOVA or mixed effect model (when missing values are present) with Šidák’s multiple comparison post-hoc test was used for comparing BW and echocardiographic variables that are normally distributed at 16–17, 20–21, and 24–25 weeks of age. The Mann–Whitney test was performed to compare histological and plasma cardiac biomarker data at 16–17 or at 24–25 weeks of age. Values of *p* < 0.05 were considered statistically significant (GraphPad Prism, Version 9.5).

## 5. Conclusions

Although the cardiac abnormalities of D2-*mdx* mice do not fully correlate with those described in DMD patients, we demonstrated that LV dysfunction was mild to moderate in both young and adult D2-*mdx* mice compared to age-matched DBA/2J WT mice. These results are in line with (1) the increase in hs-cTnT plasma concentration and normal NT-proANP levels and (2) the low percentage of interstitial fibrosis in both ventricles in 16–17- and 24–25-week-old mice. However, RV structural and functional parameters are significantly compromised. Furthermore, the D2-*mdx* mice showed greater deposition of epicardial fibrotic tissue in the RV than in their parent strain. Although there is evidence of damage in the LV, further long-term studies are needed to evaluate whether this mouse model, which displays a more severe dystrophic phenotype, can also be considered a model of DMD cardiomyopathy.

## Figures and Tables

**Figure 1 ijms-24-11805-f001:**
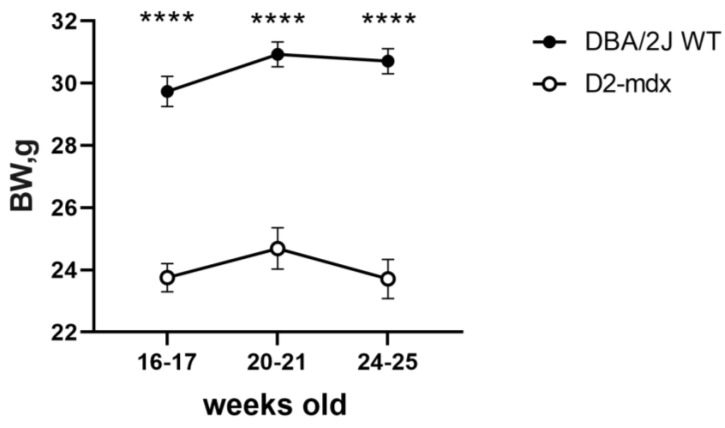
BW evolution in mice from 16–17 to 24–25 weeks of age. Data are the mean ± SEM, *n* = 10 for each group. Mixed-effects model; *p* < 0.0001; Šídák’s multiple comparisons post-hoc test; **** *p* < 0.0001 vs. DBA/2J WT.

**Figure 2 ijms-24-11805-f002:**
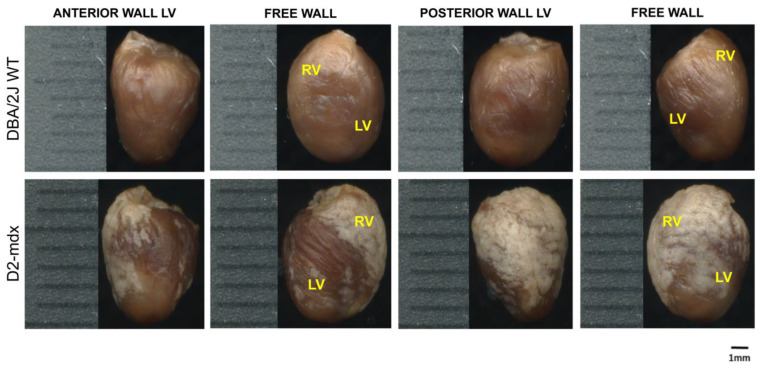
Epicardial fibrocalcinosis of the right and left ventricles in 16–17-week-old mice. Representative digitalized images of the anterior and posterior sides of formalin-fixed DBA/2J WT and D2-*mdx* mouse hearts, acquired by a high-resolution scanner in 16–17-week-old mice. Right ventricle (RV); left ventricle (LV). Scale bar = 1 mm.

**Figure 3 ijms-24-11805-f003:**
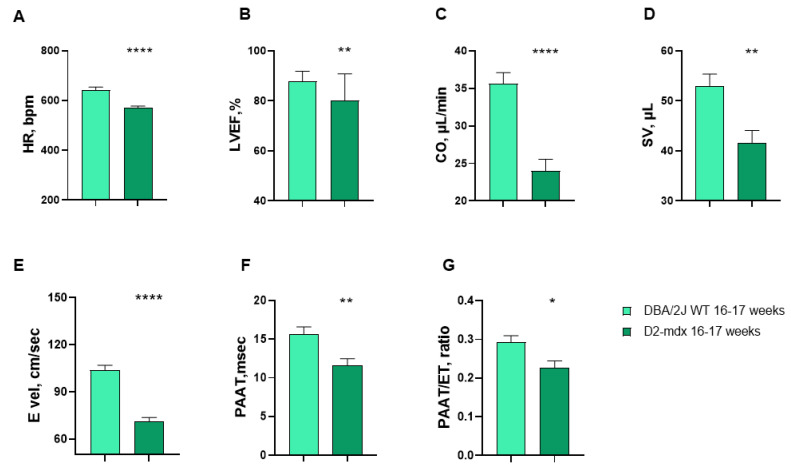
Heart rate, left ventricular echocardiographic variables, and pulmonary vascular resistance parameters evaluated in 16–17-week-old mice. Data are the mean ± SEM, *n* = 20 for each group. (**A**) Heart rate (HR); (**B**) left ventricular ejection fraction (LVEF); (**C**) cardiac output (CO); (**D**) stroke volume (SV); (**E**) mitral E inflow velocity (E vel); (**F**) pulmonary artery acceleration time (PAAT); (**G**) PAAT on pulmonary artery ejection time (ET) index (PAAT/ET). * *p* < 0.05, ** *p* < 0.01, and **** *p* < 0.0001 vs. DBA/2J WT mice, unpaired *t*-test.

**Figure 4 ijms-24-11805-f004:**
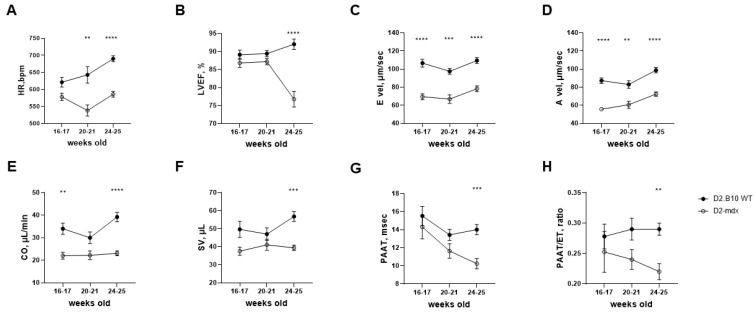
Heart rate, left ventricular echocardiographic variables, and pulmonary vascular resistance parameters evaluated in 8 weeks of follow-up (from 16–17- to 24–25-week-old mice). Data are the mean ± SEM, *n* = 10 for each group. Two-way ANOVA or mixed-effects model. (**A**) Heart rate (HR); (**B**) left ventricular ejection fraction (LVEF); (**C**) mitral E inflow velocity (E vel); (**D**) mitral A inflow velocity (A vel); (**E**) left ventricular cardiac output (CO); (**F**) left ventricular stroke volume (SV); (**G**) pulmonary artery acceleration time (PAAT); (**H**) PAAT on pulmonary artery ejection time index (PAAT/ET). Šídák’s multiple comparisons post-hoc test; ** *p* < 0.01, *** *p* < 0.001, and **** *p* < 0.0001 vs. DBA/2J WT.

**Figure 5 ijms-24-11805-f005:**
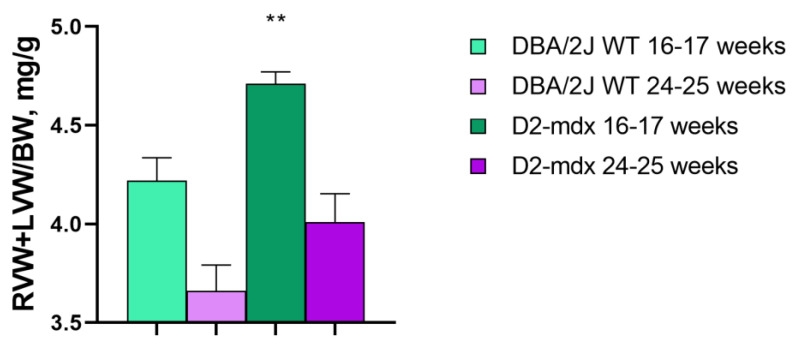
Right ventricle weight + left ventricle weight normalized by BW in 16–17- and 24–25-week-old mice. Data are the mean ± SEM, *n* = 10 for each group. Right ventricle weight (RVW); left ventricle weight (LVW). ** *p* < 0.01 vs. 16–17-week-old DBA/2J WT mice, unpaired *t*-test.

**Figure 6 ijms-24-11805-f006:**
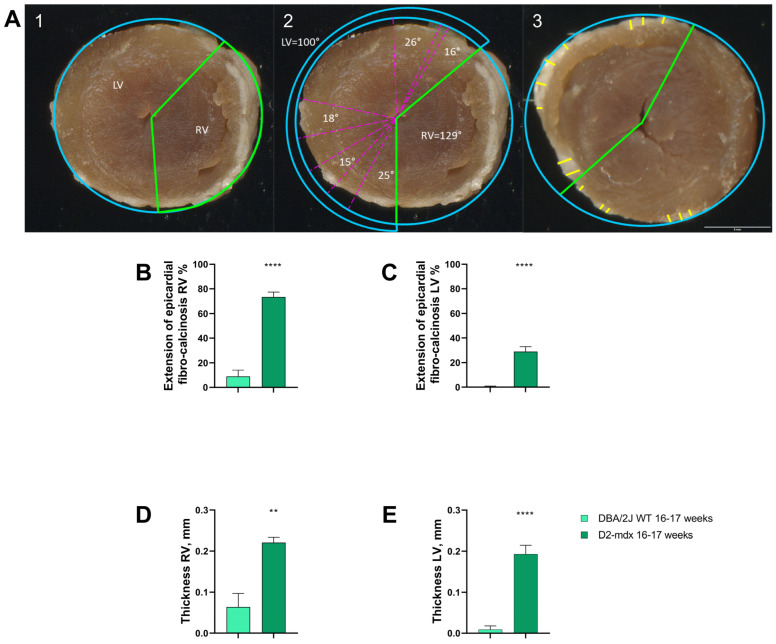
Extension and thickness of epicardial fibrocalcinosis areas (crusts) in the 16–17-week-old D2-*mdx* mice. (**A**) Angular delimitation of the left ventricle (LV; light blue line) and right ventricle (RV; green line) (**1**); example of LV and RV fibrocalcinosis angular extension in the D2-*mdx* mouse, with each angle delimited by pink segmented lines (**2**); example of LV and RV fibrocalcinosis thickness yellow lines) in the D2-*mdx* mouse (**3**). Data are the mean ± SEM, *n* = 10 for each group. (**B**,**C**) RV and LV epicardial fibrocalcinosis extensions, respectively; (**D**,**E**) RV and LV epicardial fibrocalcinosis thicknesses, respectively. ** *p* < 0.01 and **** *p* < 0.0001 vs. DBA/2J WT, Mann–Whitney test.

**Figure 7 ijms-24-11805-f007:**
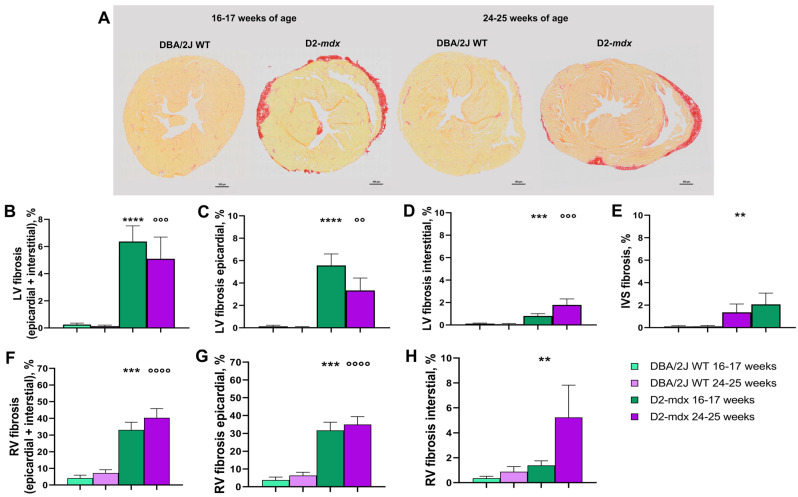
Fibrosis percentage in the right and left ventricular wall and interventricular septum in 16–17- and 24–25-week-old mice. (**A**) Fibrosis distribution in the hearts of the 16–17- and 24–25-week-old D2-*mdx* mice. Sections were stained by Sirius Red 0.1%, magnification 200×, scale bars = 500 µm. (**B**) LV epicardial plus interstitial fibrosis; (**C**) LV epicardial fibrosis; (**D**) LV interstitial fibrosis; (**E**) IVS fibrosis; (**F**) RV epicardial plus interstitial fibrosis; (**G**) RV epicardial fibrosis; (**H**) RV interstitial fibrosis. Data from Sirius Red staining analysis. Data are the mean ± SEM, *n* = 10 for each group. Left ventricle (LV); intraventricular septum (IVS); right ventricle (RV). ** *p* < 0.01, *** *p* < 0.001 and **** *p* < 0.0001 vs. 16–17-week-old DBA/2J WT mice. °° *p* < 0.01, °°° *p* < 0.001, and °°°° *p* < 0.0001 vs. 24–25-week-old DBA/2J WT mice, Mann–Whitney test.

**Figure 8 ijms-24-11805-f008:**
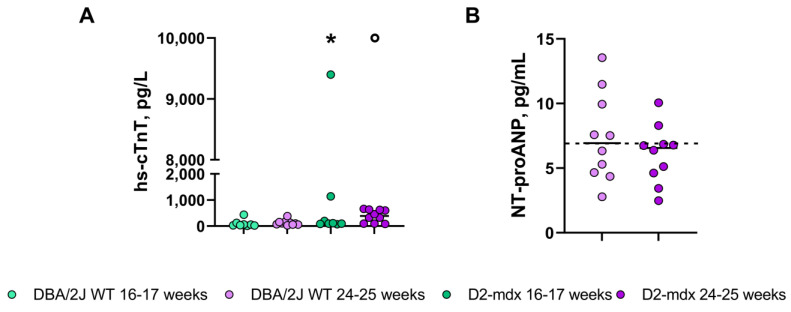
Plasma concentration of cardiac biomarkers in the 16–17- and at 24–25-week-old mice. Data expressed as the median [Q1–Q3], *n* = 10 for each group. (**A**) High-sensitivity cardiac troponin T (Hs-cTnT); (**B**) N-terminal pro-atrial natriuretic peptide (NT-proANP). Segmented line indicates cutoff for normality values. * *p* < 0.05 vs. 16–17-week-old DBA/2J WT mice; ° *p* < 0.05 vs. 24–25-week-old DBA/2J WT mice, Mann–Whitney test.

**Figure 9 ijms-24-11805-f009:**
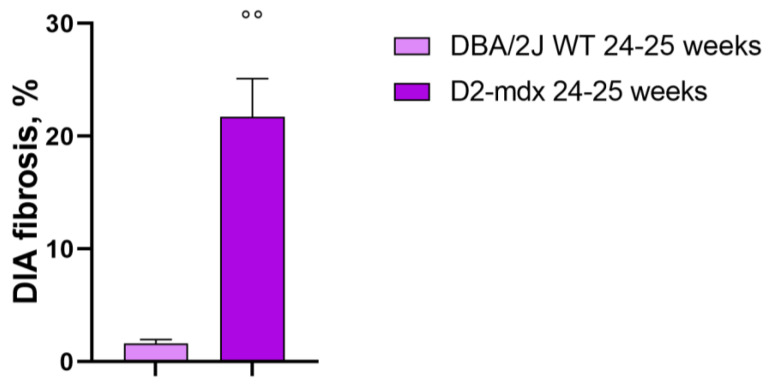
Fibrosis deposition in the diaphragm muscle in 24–25-week-old mice. Data are the mean ± SEM, *n* = 5 for each group. Diaphragm (DIA). °° *p* < 0.01 vs. the DBA/2J WT mice, Mann–Whitney test.

**Figure 10 ijms-24-11805-f010:**
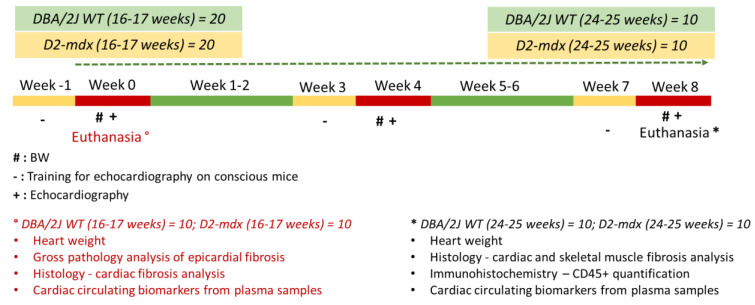
Study flow-chart. BW, body weight.

## Data Availability

The datasets used and/or analyzed during the current study are available from the corresponding author upon reasonable request.

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
