# Peer review of "Characterization of the Cardiac Structure and Function of Conscious D2.B10-Dmdmdx/J (D2-mdx) mice from 16–17 to 24–25 Weeks of Age"

_ijms, 2023, doi:10.3390/ijms241411805_

Round 1

Reviewer 1 Report

The objectives of this study were to follow the evolutive phenotype of the heart in D2-mdx mice to understand if this is a good model for DMD pathology. The authors cover a large panel of experiments from echocardiography, serology, gross anatomy and histology on mice aged from 16 weeks until 25 weeks and compare WT and mdx mice. They conclude that D2-mdx mice present a mild to moderate cardiac phenotype that is more pronounced in adult compared to young mice. However, this is not completely recapitulating the human phenotype as fibrosis is mainly found in the right ventricle in the mouse while in man this is the Left ventricle that is affected.

Overall, this analysis is mainly descriptive.

Major points:

- In the results part, it is unclear of what represents LV and RV extension presented in S1?  

- There are some discrepancies between the text and Figure 5: Heart weight similar between mdx and WT at 16weeks in the figure while the text indicates “RV+LV weight on BW was higher in the D2-mdx group when compared to DBA/2J WT mice starting from 16/17 up to 24/25 weeks of age”. Why the percentage is in negative while the authors are talking about an increase ?

- What do you mean by crust thickness while you are looking at heart anatomy ?

- The results concerning the quantification of fibrosis are unclear ? What do you mean by epicardial and interstitial fibrosis ? how do you explain less fibrosis in older mdx mice ? Images at 16 weeks are required to be able evaluate this. What is the difference between crusts and fibrosis ?

- In general, I recommend to add representative images of the data in the figure to help the reading instead of a figure composed only by graphs.

- L260 What do you mean by conscious mice ? They were not anesthetized. What is the advantage of echo in conscious mice ? what differences compared with anesthetized mice ? The procedure is not well described in the material and methods and this is not indicated in the results ?

Minor points

For clarity reason, it is best to add the real age of the mice in the X-axis of each graph instead of 0-4-8 weeks of the follow-up.

Line247: add “in adult” as the young mdx do not present higher serologic level of cTnT except for two who may have already developed the pathology.

Line 499: “Example of angular extension and thickness measurements.” Precise the extension in supplementary figure 1 ?

Reviewer 2 Report

The authors present a complementary series of descriptive analyses of cardiac structure and function in a murine model of DMD. The authors find compromised cardiac function and structure and suggest this model can be used in further understanding the pathology and possible treatment effects on the heart in this model. The experiments are well conducted and the paper is well written. This will be of interest to those in the DMD field and seems a natural contribution to the special issue.

The work is novel and addresses one of many gaps in DMD literature, particularly as they relate to inability of mouse models to fully model human DMD symptoms. In this instance cardiac defects.   It adds new data in a newer model and suggests this newer model might be a better model for cardiac defects.   The figures are of a good quality and present the data that support the author's conclusions.

Round 2

Reviewer 1 Report

No more comments, you have answered to all of them.